A tropomyosin receptor kinase family protein, NTRK2 is a potential predictive biomarker for lung adenocarcinoma

Wang Xiang 1 2
http://orcid.org/0000-0003-2047-883X Xu Zhijie 3
Chen Xi 1 2
Ren Xinxin 4
Wei Jie 1 2
Zhou Shuyi 5
Yang Xue 1 2
Zeng Shuangshuang 1 2
Qian Long 1 2
Wu Geting 3
Gong Zhicheng 1 2
Yan Yuanliang 1 2 yanyuanliang@csu.edu.cn
1 Department of Pharmacy, Xiangya Hospital, Central South University , Changsha, Hunan , China
2 National Clinical Research Center for Geriatric Disorders, Xiangya Hospital, Central South University , Changsha, Hunan , China
3 Department of Pathology, Xiangya Hospital, Central South University , Changsha, Hunan , China
4 Center for Molecular Medicine, Xiangya Hospital, Key Laboratory of Molecular Radiation Oncology of Hunan Province, Central South University , Changsha, Hunan , China
5 Department of General Surgery, Hunan Provincial People’s Hospital Xingsha Branch (People’s Hospital of Changsha County) , Changsha, Hunan , China
Khan Imran
Electronic publication date: 2019 Jun 17
Publication date: 2019
Volume: 7
Electronic Location ID: e7125
Received 2019 Feb 11; Accepted 2019 May 14
Copyright: © 2019 Wang et al.
Copyright year: 2019
Copyright holder: Wang et al.
License: This is an open access article distributed under the terms of the Creative Commons Attribution License, which permits unrestricted use, distribution, reproduction and adaptation in any medium and for any purpose provided that it is properly attributed. For attribution, the original author(s), title, publication source (PeerJ) and either DOI or URL of the article must be cited.
License URL: https://creativecommons.org/licenses/by/4.0/

Keywords: NTRK2, Lung adenocarcinoma, Expression, Diagnosis, Prognosis

Funding: National Natural Science Foundation of China 81803035, 81703036 and 81572946 Natural Science Foundation of Hunan Province 2019JJ50932 China Postdoctoral Science Foundation 2017M610510 Open-End Fund for the Valuable and Precision Instruments of Central South University CSUZC201836 Youth Fund of Xiangya Hospital 2017Q17 Postdoctoral Science Foundation of Central South University 185702 The study was supported by the grants from National Natural Science Foundation of China (81803035, 81703036, 81572946), Natural Science Foundation of Hunan Province (No. 2019JJ50932), China Postdoctoral Science Foundation (2017M610510), Open-End Fund for the Valuable and Precision Instruments of Central South University (CSUZC201836), Youth Fund of Xiangya Hospital (2017Q17) and Postdoctoral Science Foundation of Central South University (185702). The funders had no role in study design, data collection and analysis, decision to publish, or preparation of the manuscript.

==============================
Neurotrophic receptor tyrosine kinase 2 (NTRK2) is a member of the tropomyosin receptor kinase family associated with the tumor development. However, the detailed function of NTRK2 in lung cancer, especially in lung adenocarcinoma (LUAD), is still not fully understood. Here, we investigated the effects of NTRK2 on LUAD biology. Through analyzing bioinformatics data derived from several databases, such as Oncomine, Gene Expression Profiling Interactive Analysis and UALCAN, we found that NTRK2 expression was significantly decreased in LUAD tissues. Clinical data acquired from Wanderer database, which is linked to The Cancer Genome Atlas database, demonstrated that the expression and methylation site of NTRK2 were significantly related to the clinical characteristics and prognosis of LUAD. Furthermore, NTRK2 expression was increased remarkably after treatment with the protein kinase B (AKT) inhibitor MK2206 and the anticancer agent actinomycin D. Functional enrichment analysis of NTRK2-associated coexpression genes was further conducted. Together, our results suggested that downregulated NTRK2 might be used in the diagnostic and prognostic evaluation of LUAD patients, or as a potential therapeutic target for the treatment of LUAD.

Introduction

Lung adenocarcinoma (LUAD) is the most frequent subtype of lung cancer, with incidence and mortality rates rising in both Western and Asian countries (Yan et al., 2019). Because of late diagnoses, the 5-year overall survival (OS) rate LUAD varies from 4 to 17% in line with the differences of stage and region, which is still very poor (Yan et al., 2018). At present, there is still no effective early diagnosis method for patients to receive timely treatment (Zheng et al., 2018). Therefore, it is necessary to search for novel target molecules for improving the early diagnosis and treatment of LUAD.

Previous studies have found a strong link between neurotrophic receptor tyrosine kinase 2 (NTRK2) and psychiatric disorders, such as schizophrenia (Spalek et al., 2016). Recent research advancement in the field revealed the relationship between NTRK2 and cancer biology. According to the ceRNA network, Gao et al. (2019) found that NTRK2 is related to the prognosis of invasive breast cancer. Through constructing the coexpression modules by WGCNA, NTRK2 was proposed to play a key role in the recurrence of uveal melanoma (Wan et al., 2018). Ni et al. (2017) demonstrated that activated NTRK2 alleles, especially the human tumor-associated QKI-NTRK2 fusion, could function together with Ink4a/Arf loss to promote astrocytoma formation. Furthermore, a recent study found that the interaction between differentiated glioblastoma cells and stem-like tumor cells via BDNF-NTRK2-VGF paracrine signaling accelerates tumor growth (Wang et al., 2018b). Nevertheless, there were few investigations about the relationship between NTRK2 and lung cancer, particularly LUAD, so the effects and mechanisms of NTRK2 in LUAD require further research.

The purpose of our study was to evaluate the role and mechanism of NTRK2 in human LUAD. Through bioinformatics data analysis, NTRK2 was found to be significantly downregulated in LUAD tissues. In addition, the expression level and methylation site of NTRK2 were notably correlated with clinical characteristics and prognosis. Moreover, based on the two datasets GSE6400 and GSE54293 from Gene Expression Omnibus (GEO), we observed the high levels of NTRK2 in the anticancer treatment group, indicating that NTRK2 could be used as a biomarker in evaluating clinical efficacy. In addition, Gene Ontology (GO) enrichment and Kyoto Encyclopedia of Genes and Genomes (KEGG) (Kanehisa & Goto, 2000) analysis of NTRK2-associated coexpression genes further indicated that NTRK2 played an important part in LUAD treatment.

Materials and Methods

Data acquisition and reanalysis using different bioinformatics tools

The relevant bioinformatics data analysis of NTRK2 was obtained from several bioinformatics web resources, which were summarized in Table S1. The flow diagram of NTRK2 screen is shown in Fig. S1.

Oncomine is a cancer microarray or high-throughput sequencing data-mining platform, from which we can get gene expression signatures in human cancer tissues and cells (Rhodes et al., 2004). The data in Oncomine could be also linked into other public databases, such as GEO and The Cancer Genome Atlas (TCGA) (Hutter & Zenklusen, 2018). We conducted the comparison of NTRK2 expression across eight analyses between the LUAD and normal tissues. Additionally, Gene Expression Profiling Interactive Analysis (GEPIA) (Tang et al., 2017a), GE-mini (Tang et al., 2017b), Cancer RNA-Seq Nexus (CRN) (Li et al., 2016) and UALCAN (Chandrashekar et al., 2017), four additional cancer microarray or high-throughput sequencing data-mining databases, were employed to verify the results.

Wanderer is an interactive viewer, providing gene expression and DNA methylation data in human cancer (Diez-Villanueva, Mallona & Peinado, 2015), which enables us to screen for the possible methylation sites in the NTRK2 DNA sequence and to analyze the correlation between clinical characteristic of LUAD patients and NTRK2 expression and methylation sites. For the prognostic analysis, the Kaplan–Meier Plotter, a tool that can be used to assess the effect of genes on survival (Wang et al., 2018a), was utilized to describe the relationship between NTRK2 expression level, overall survival time (OS) and post-progression survival time (PPS). Further, the association between NTRK2 expression and disease-free survival (DFS) was completed through the GEPIA database.

Two datasets of the treatment-related transcriptome microarray, GSE6400 (Wang et al., 2007) and GSE54293 (Denisova et al., 2014), were acquired from the GEO database (Barrett & Edgar, 2008). Subsequently, the effects of NTRK2 expression on the chemotherapy for LUAD were analyzed.

The expression and methylation of NTRK2 correlation analysis were implemented by MethHC, which provided the information of DNA methylation and gene expression in human cancer (Huang et al., 2015). For the relevance between the disease prognosis and the methylation sites of NTRK2, the MethSurv tool was employed (Modhukur et al., 2018).

Using the cBioportal web tool (Gao et al., 2013), genes coexpressed with NTRK2 in LUAD were downloaded. Then, the STRING database (Szklarczyk et al., 2017) and Cytoscape software (Reimand et al., 2019) were used to complete the protein–protein interaction (PPI) network of these coexpression genes. Then, we utilized the DAVID bioinformatics resource (Huang, Sherman & Lempicki, 2009) to conduct the GO and KEGG pathway analysis of NTRK2 coexpression genes in LUAD samples. The web tools of WebGestalt (Wang et al., 2017) and PATHVIEW (Luo et al., 2017) were used for building a graphic.

Statistical analyses

The statistical tests were performed using SPSS 12.0 software (IBM Analytics). The results were expressed as the mean ± SD. Student t test, one-way ANOVA and K independent samples test were performed when appropriate. P < 0.05 was considered statistically significant.

Results

NTRK2 is downregulated in LUAD tissues

The NTRK family consists of three members, NTRK1, NTRK2 and NTRK3. Through the bioinformatics analysis of databases, we evaluated the transcriptional levels of NTRK family members in LUAD. First, we used the Oncomine database to observe the expression of NTRK1, NTRK2 and NTRK3 in eight LUAD datasets (Beer et al., 2002; Bhattacharjee et al., 2001; Hou et al., 2010; Landi et al., 2008; Okayama et al., 2012; Selamat et al., 2012; Stearman et al., 2005; Su et al., 2007). The results showed that NTRK2 had significantly lower expression in LUAD through the comparison among nine datasets, whereas NTRK1 and NTRK3 showed no statistical significance (Fig. 1A). Therefore, NTRK2 was chosen as the research target. To verify the trend, we examined the NTRK2 expression in LUAD by GEPIA and GE-mini, and we discovered the NTRK2 expression was clearly reduced in LUAD compared with the normal tissues (Figs. 1B and 1C). In addition, the heatmap from CRN database further indicated the low expression of NTRK2 in LUAD tissues (Fig. 1D). Next, given some activated oncogenes, such as Erb-B2 receptor tyrosine kinase 2 (ERBB2) and MET, have been demonstrated the driver roles in LUAD (The Cancer Genome Atlas Research Network, 2014), we want to evaluate the association between NTRK2 and these oncogenes. The data from UALCAN revealed the significantly downregulated NTRK2 (P < 0.01), upregulated ERBB2 (P < 0.01) and upregulated MET (P < 0.01) in LUAD tissues (Fig. S2A). Spearman correlation analysis showed the negative association between the expression of NTRK2 and ERBB2 or MET (Fig. S2B). Taken together, all of the above data suggested that the decreased expression of NTRK2 contributed to LUAD tumorigenesis, supporting its tumor-inhibiting function in LUAD.

Figure 1 Analysis of NTRK2 expression levels in LUAD tissues.

(A) The comparison of the messenger RNA (mRNA) expression of NTRK (NTRK1, NTRK2 and NTRK3) among eight datasets by comparing the surrounding normal lung tissues and LUAD. (B–D) The mRNA expression of NTRK2 was evaluated from the database GEPIA, GE-mini and CRN, respectively.

NTRK2 expression is associated with the clinical characteristics of LUAD patients

After determining the expression of NTRK2 in LUAD, we further analyzed the correlation between the NTRK2 expression level and the clinical characteristics of patients. Using the Wanderer database, we obtained a series of clinical data, and a summary of clinical characteristic parameters is provided in Table 1. As shown in this table, NTRK2 expression was significantly associated with gender (P = 0.007), pathologic T (P = 0.021), pathologic M (P = 0.006) and age (P = 0.036). Then, the Kaplan–Meier Plotter tool was used to evaluate the effects of NTRK2 expression on OS and PPS, confirming that the downregulated of NTRK2 expression was significantly related to shorter OS (P = 0.00029) (Fig. 2A) and PPS (P = 0.021) (Fig. 2B). Furthermore, we found that low NTRK2 expression was associated with RFS (P = 0.012) through using the GEPIA database (Fig. 2C). In conclusion, NTRK2 could be as a potential biomarker both for diagnosis and prognosis.

Figure 2 The effects of NTRK2 expression on prognosis in LUAD patients.

(A–B) The relationship between NTRK2 expression and OS and PPS, described by Kaplan–Meier Plotter. (C) The association between NTRK2 expression and RFS within the GEPIA database.

Table 1 The correlation between clinical characteristic parameters and the expression of NTRK2 in LUAD.

Variables	Number	Mean ± SD	P	
Gender			0.007	
 Male	179	5.25 ± 1.90		
 Female	212	5.78 ± 1.95		
Radiation therapy			0.640	
 Yes	6	5.12 ± 1.31		
 No	89	5.48 ± 1.82		
Kras mutation found			0.454	
 Yes	14	5.48 ± 1.89		
 No	34	5.88 ± 1.54		
Pathologic T			0.021	
 T1/T1a/T1b	122	6.01 ± 1.94		
 T2/T2a/T2b	218	5.35 ± 2.01		
 T3	34	5.29 ± 1.43		
 T4	15	5.09 ± 1.52		
 TX	2	4.36 ± 1.56		
Pathologic N			0.875	
 N0	252	5.52 ± 1.86		
 N1	71	5.59 ± 1.89		
 N2	61	5.55 ± 2.32		
 NX	5	4.85 ± 2.40		
Pathologic M			0.006	
 M0	255	5.38 ± 1.85		
 M1/M1a/M1b	16	4.86 ± 2.20		
 MX	117	5.99 ± 2.04		
Pathologic stage			0.471	
 Stage I/IA/IB	211	5.63 ± 1.89		
 Stage IIA/IIB	94	5.45 ± 1.78		
 Stage IIIA/IIIB	68	5.52 ± 2.25		
 Stage IV	17	4.89 ± 2.14		
Race			0.758	
 White	314	5.60 ± 1.92		
 Black or African American	23	5.41 ± 2.26		
 Asian	5	5.07 ± 1.12		
Tobacco smoking history			0.097	
 Current reformed smoker for > 15 years	94	5.84 ± 2.06		
 Current reformed smoker for < or = 15 years	131	5.38 ± 1.95		
 Current reformed smoker, duration not specified	2	5.15 ± 1.34		
 Lifelong non-smoker	61	5.91 ± 1.75		
 Current smoker	91	5.21 ± 1.97		
Age at initial pathologic diagnosis			0.036	
 ≤60	125	5.25 ± 1.86		
 >60	248	5.69 ± 1.94		
EGFR mutation result			0.303	
 Exon 19 deletion	7	5.07 ± 1.26		
 L858R	3	6.47 ± 1.00		
 Other	9	5.94 ± 1.58		

The roles of NTRK2 in LUAD therapies

For the purpose of identifying the exact function of NTRK2 in LUAD chemotherapy, two treatment-related transcriptome microarray datasets, GSE6400 and GSE54293, were obtained from the GEO database. Previous studies have demonstrated that actinomycin D (Bai et al., 2019) and MK2206 (Dai et al., 2017) were two promising antitumor drugs. In the GSE6400 dataset, we discovered that the expression of NTRK2 was apparently higher in the actinomycin D treatment group than in the mannitol-control group (P = 0.008) (Fig. 3A). In addition, for the GSE54293 dataset, the AKT inhibitor MK2206 could enhance the NTRK2 expression levels significantly (P = 0.009) (Fig. 3B). Collectively, the findings observed above suggested that NTRK2 might enhance the response of cancer cells to the chemotherapeutics.

Figure 3 The influence of NTRK2 on the therapeutic response of LUAD patients.

(A) The GSE6400 dataset acquired from the GEO database was employed to estimate the impacts of NTRK2 expression on LUAD therapy both in the actinomycin D treatment group and the mannitol-control group. (B) In the treatment-related microarray GSE54293 dataset, the influence of NTRK2 expression on AKT inhibitor MK2206 treatment was evaluated.

The relationship between NTRK2 methylation and the clinical characteristics of LUAD patients

It is well-known that there is a negative correlation between DNA methylation and gene expression (Shi et al., 2017; Zhou et al., 2019). From the MethHC database, we observed that global NTRK2 methylation was significantly higher in LUAD samples compared with normal samples (P < 0.005) (Fig. 4A) and was negatively related to its expression (P = 0.000) (Fig. 4B), which gives further support for the low expression of NTRK2 in LUAD. Subsequently, the methylation site cg03628748 was screened out of the data (P = 4.35E-12) (Table S2) acquired from the Wanderer database. Then, the relationship between cg03628748 and the clinical characteristics of LUAD patients was examined, and results showed that cg03628748 was significantly related to Kras mutation (P = 0.038) and pathologic T (P = 0.000) (Table 2). Moreover, there was a significant negative correlation between higher methylation value of cg03628748 and shorter OS in LUAD patients (P = 0.034), which was analyzed by using the web tool of MethSurv (Fig. 4C).

Figure 4 The relationship between NTRK2 methylation and the clinical characteristics of LUAD patients.

(A) Global NTRK2 methylation in LUAD samples compared with the normal samples analyzed by MethHC database. (B) The association between global NTRK2 methylation and its expression in LUAD samples using the MethHC database. (C) The impact of the methylation site cg03628748 in NTRK2 on OS in LUAD patients as analyzed by the MethSurv web tool.

Table 2 The correlation between clinical characteristics of patients and the methylation site cg03628748 in NTRK2 in LUAD.

Variables	Number	Mean ± SD	P	
Gender			0.123	
 Male	189	0.32 ± 0.14		
 Female	219	0.30 ± 0.14		
Radiation therapy			0.112	
 Yes	7	0.22 ± 0.097		
 No	96	0.31 ± 0.14		
Kras mutation found			0.038	
 Yes	16	0.38 ± 0.18		
 No	34	0.28 ± 0.13		
Pathologic T			0.000	
 T1/T1a/T1b	127	0.26 ± 0.11		
 T2/T2a/T2b	227	0.32 ± 0.14		
 T3	36	0.35 ± 0.16		
 T4	15	0.30 ± 0.15		
 TX	3	0.23 ± 0.17		
Pathologic N			0.464	
 N0	261	0.31 ± 0.14		
 N1	75	0.30 ± 0.13		
 N2	62	0.30 ± 0.14		
 NX	8	0.24 ± 0.11		
Pathologic M			0.183	
 M0	264	0.31 ± 0.14		
 M1/M1a/M1b	17	0.27 ± 0.16		
 MX	123	0.29 ± 0.13		
Pathologic stage			0.746	
 Stage I/IA/IB	218	0.31 ± 0.14		
 Stage IIA/IIB	102	0.31 ± 0.13		
 Stage IIIA/IIIB	68	0.31 ± 0.14		
 Stage IV	19	0.27 ± 0.16		
Race			0.214	
 White	325	0.30 ± 0.13		
 Black or African American	29	0.27 ± 0.12		
 Asian	5	0.37 ± 0.14		
Tobacco smoking history			0.075	
 Current reformed smoker for > 15 years	101	0.31 ± 0.15		
 Current reformed smoker for < or = 15 years	135	0.32 ± 0.14		
 Current reformed smoker, duration not specified	2	0.39 ± 0.022		
 Lifelong non-smoker	62	0.26 ± 0.12		
 Current smoker	96	0.31 ± 0.14		
Age at initial pathologic diagnosis			0.644	
 ≤60	131	0.30 ± 0.14		
 >60	259	0.31 ± 0.13		
Residual tumor			0.542	
 RX	16	0.31 ± 0.15		
 R0	271	0.31 ± 0.14		
 R1	10	0.26 ± 0.093		
EGFR mutation result			0.082	
 Exon 19 deletion	7	0.18 ± 0.063		
 L858R	3	0.28 ± 0.17		
 Other	9	0.33 ± 0.13		

Functional enrichment analysis of NTRK2-associated coexpression genes

Using the cBioPortal database, 15,146 genes that were notably coexpressed with NTRK2 in the LUAD samples were acquired. The volcano plot was established for exhibiting between the altered and unaltered NTRK2 expression group (Fig. 5A). Next, we singled out 219 NTRK2-associated codifferentially expressed genes (co-DEGs) with the criteria of P value < 0.05 and |log Ratio| ≥ 2 (Table S3). Then, a PPT network of the co-DEGs was performed by using the STRING database and Cytoscape software (Fig. 5B). For the purpose of comprehending the biological function for these co-DEGs, GO and KEGG analyses were conducted by the WebGestalt and PATHVIEW web tools, respectively. The biological processes showed that these co-DEGs were mainly connected with biological regulation and metabolic processes (Fig. 5C). For the analysis of cellular components, the coexpression genes were mainly localized on cell membranes (Fig. 5D). For molecular function, protein binding was primarily enriched for these coexpression genes (Fig. 5E). Furthermore, the KEGG pathway demonstrated that these genes were involved in the process of xenobiotics and drug metabolism by cytochrome P450 (Table S4).

Figure 5 Functional enrichment analysis of NTRK2-associated co-DEGs in LUAD.

(A) The coexpression genes of NTRK2 were shown as volcano plot. (B) The PPI network of NTRK2-associated co-DEGs as completed by the STRING and Cytoscape software. (C–E) The GO analysis of NTRK2 associated co-DEGs including biological processes, cellular components and molecular function.

Discussion

This is the first study which presents comprehensive bioinformatic analysis of different public datasets that NTRK2 was identified as anti-oncogene in LUAD and could be used as a potential biomarker. Using the TCGA data from several databases, we found that NTRK2 expression was markedly decreased in LUAD tissues. The patients with downregulated NTRK2 expression and higher methylation values often had shorter OS, PPS and RFS.

Neurotrophic receptor tyrosine kinase 2 belongs to the NTRK family and has been previously shown to have an important impact on the development of the nervous system (Cocco, Scaltriti & Drilon, 2018). However, recent studies have demonstrated the possible role of NTRK2 in the development of cancer. Neurotrophic receptor tyrosine kinase 2 activation cooperates with PTEN deficiency through the activation of both the JAK–STAT3 and PI3K-AKT pathways to induce aggressiveness, resistance to current therapies and poor prognosis of T-cell acute lymphoblastic leukemia (Yuzugullu et al., 2016). Currently, NTRK fusion mutations have been reported to associate with oncogenic activation in various signaling pathways, such as AKT and MAPK, across multiple tumors (Stransky et al., 2014). Moreover, NTRK fusions were connected with poor survival in lung cancers (Rolfo & Raez, 2017). Interestingly, the reports seemed contrary to our results; this phenomenon might be explained by following reasons. First, it is known that different diseases or subtypes of tumors have diverse pathological states, which can change genes, functions. On the other hand, the structure, constitution and condition of genes may transformed, such as gene mutation, accompany with gene fusions. Furthermore, NTRK fusions are thought to occur at a low frequency across multiple tumor types (Vaishnavi, Le & Doebele, 2015). Additionally, although NTRK fusions were observed in rare cancer types, such as congenital infantile fibrosarcoma and secretory breast carcinoma, the occurrence in common cancers has been largely unexplored (Qaddoumi et al., 2016). Additionally, the difference in results might be on account of study designs or different patient populations, indicating international, multicenter randomized controlled, clinical research is needed for further study.

In the present study, GO and KEGG pathway analyses indicated that genes coexpressed with NTRK2 were mainly enriched in the processes of xenobiotics and drug metabolism. Moreover, NTRK2 expression was much higher in drug therapy groups in both the GSE6400 and GSE54293 datasets. Therefore, up-regulating NTRK2 expression to promote drug metabolism might be the mechanism that explains this phenomenon.

Nevertheless, there were several limitations to our study. First, the flow chart of analysis on the roles of NTRK2 in LUAD tumorigenesis was not strong enough, and should be further verified externally in diverse cohorts. Additionally, further validation of the roles of NTRK2 in multicenter clinical trials and prospective research is required. For the TCGA database, the included ethnicities were primarily white and black, and more studies are needed to confirm whether the findings are appropriate for other ethnic groups. Furthermore, more prognostic variables must be included to improve performance.

Conclusion

In conclusion, our study illustrated that NTRK2 was a putative cancer suppressor gene and could serve as a promising biomarker in tumorigenesis and treatment of LUAD patients. Furthermore, DNA hypermethylation has been demonstrated to be one of the mechanisms for the low-expressed NTRK2 in LADC. Understanding its detailed function and mechanisms in LUAD biological processes would provide promising insights for the prognostic and therapeutic value.

Supplemental Information

Supplemental Information 1 Supplemental figures and tables.

Figure S1. A Flow chart of analysis on the roles of NTRK2 in LUAD tumorigenesis.

Figure S2. The negative association between the expression of NTRK2 and ERBB2 or MET in LUAD.

Table S1: The main bioinformatics tools applied to analyze the role of NTRK2 in LUAD biological processes.

Table S2: The methylation values of CpG islands in NTRK2.

Table S3: The NTRK2-associated co-DEGs in LUAD.

Table S4: The KEGG pathway of NTRK2-associated co-DEGs.

Click here for additional data file.

We thank Elsevier’s English Language Editing Service for assistance with the language editing.

Additional Information and Declarations

Competing Interests

Author Contributions

Data Availability

The authors declare that they have no competing interests.

Xiang Wang performed the experiments, analyzed the data, prepared figures and/or tables.

Zhijie Xu conceived and designed the experiments, performed the experiments, analyzed the data, prepared figures and/or tables, authored or reviewed drafts of the paper, approved the final draft.

Xi Chen contributed reagents/materials/analysis tools.

Xinxin Ren analyzed the data, contributed reagents/materials/analysis tools, prepared figures and/or tables.

Jie Wei contributed reagents/materials/analysis tools.

Shuyi Zhou contributed reagents/materials/analysis tools.

Xue Yang contributed reagents/materials/analysis tools.

Shuangshuang Zeng contributed reagents/materials/analysis tools.

Long Qian contributed reagents/materials/analysis tools.

Geting Wu contributed reagents/materials/analysis tools.

Zhicheng Gong conceived and designed the experiments.

Yuanliang Yan conceived and designed the experiments, performed the experiments, authored or reviewed drafts of the paper, approved the final draft.

The following information was supplied regarding data availability:

The raw measurements are available in the Supplemental Tables. The raw data shows the main bioinformatics tools, methylation values of CpG islands in NTRK2 and NTRK2-associated codifferentially expressed genes in lung adenocarcinoma.

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
