# Peer review of "A tropomyosin receptor kinase family protein, NTRK2 is a potential predictive biomarker for lung adenocarcinoma"

_PeerJ, doi:10.7717/peerj.7125_

## Round 0.1 · original submission · Major Revisions

Both the reviewers have suggested multiple changes and have asked critical questions. Also, please modify manuscript English before resubmitting.

Reviewer 1 ·

Basic reporting

The manuscript entitled "NTRK2 is a potential predictive biomarker for lung
adenocarcinoma" by Yuanliang Yan et al. studies the data of expression profile of NTRK2 gene in lung adenocarcinoma.

Overall the manuscript is satisfactory. The findings are worth to report and will be useful in the studies mainly in understanding the NTKR2 expression profile. Literature is well referenced & relevant to the concept.

Experimental design

Not applicable.

Validity of the findings

Findings are valid, need additional controls for the study.

Additional comments

I have the following major concerns:
By considering NTRK tyrosine kinase family genes only and selecting only NTRK2 for further study appears as a biased approach. I would suggest that author should take control gene or set of genes that have regulated expression in lung adenocarcinoma, for the correlation based parallel studies and compare the data from normal and cancerous tissues.

Title of the manuscript is not a self-explanatory "NTRK2 is a potential predictive biomarker for lung adenocarcinoma". NTRK2 is a very specific abbreviated name of protein. Keeping in mind the broad audience of this (PeerJ) journal, author should expand or generalize the title, as for example “A tropomyosin receptor kinase (TRK) family protein, NTRK2 is a potential predictive biomarker for lung adenocarcinoma”.

“Neurotrophic receptor tyrosine kinase 2 (NTRK2), a member of the tropomyosin receptor kinase (TRK) family, has been found to be associated with the development of tumors.” This is passive voice statement and jargon. Active voice and direct sentences would be preferably used in all through the manuscript for e.g.
“Neurotrophic receptor tyrosine kinase 2 (NTRK2) is a member of the tropomyosin receptor kinase (TRK) family associated with the tumor development.”

Line 53-54: Currently, the relationship between NTRK2 and cancer has been found with continuous research. What is the meaning of this sentence?
Can be written as 'Recent research advancement in the field revealed the relationship between NTRK2 and cancer'

Line 63: researched to research

Line 65: ‘analyses of data’ to ‘data analysis’

Line 75: ‘methods’ to ‘tools’

Line 76: ‘analysis data’ to ‘data analysis’

Line 111-112: ‘Through the analysis of bioinformatics databases’ to ‘The bioinformatics analysis of databases’. Author is analyzing the database using bioinformatics tools and not studying the bioinformatics program itself.

Line112: “We evaluated the expression profiles of NTRK family members in LUAD.”
Authors need to explain first, which expression profile (mRNA/protein etc.) they are going to talk about.

Line 113-116: First, we used the Oncomine database to observe the expression of NTRK1, NTRK2 and NTRK3 in LUAD. The results showed that NTRK2 had significantly lower expression in LUAD through the comparison among nine datasets, whereas NTRK1 and NTRK3 showed no statistical significance (Figure 1A).”
Although, it is mentioned in figure legend, it would be appropriate that author could setup the reader accordingly. Specifically, author could have explained briefly why they are looking into Oncomine database? What does Oncomine database will provide (like RNA seq dataset, mRNA expression profile, tissues specific, cell type specific etc.,) and how oncomine database analysis will fit into their objective(s)? This journal (PeerJ) caters the broad audience and not very specific to lung cancer or kinase protein family, so author should structure their paper accordingly. This is one example I mentioned here, but the same can be done in all the section were author had used any database. Please provide the rationale, what you are looking for, and how the data analysis fit in your objective?

Texts in figures are hard to read, author should use ‘arial’ font throughout the figure.
Fig 1c: too large, please resize the figure.

Figure 2: Unreadable, please change the figure, text, axis, figure number.

Figure 4: Again same problem, please improve figure, text, axis labeling etc.,

Figure 6: This is direct use of flowchart from KEGG. It is not acceptable in current ‘as it is’ form. Author need to draw them in their own format and move the figure in supplementary section.

Combine all supplementary files in a single document.

Reviewer 2 ·

Basic reporting

Manuscript should be redrafted for English.
References should be appropriately cited.

Experimental design

The article “NTRK2 is a potential predictive biomarker for lung adenocarcinoma” described the investigation on NTRK2 down regulatory function in lung adenocarcinoma cancer using bioinformatics analysis of literature database. However, claim of this study as mentioned in introduction is to evaluate the role and mechanism of NTRK2 in human LUAD is unclear, and stronger arguments/controls connecting the results and conclusions are needed.
In general, the figures are difficult to read (especially figure 2, 5B, and 6 with low resolutions), requires appropriate citations, and figure legends as well as the relationship between the data and the proposed model is poorly explained. Additionally, it possesses significant lack of positive and negative controls or communication of controls with the appropriate data set.
I think the paper requires a major revision.

Major Concerns:
1. Kyoto Encyclopedia of Genes and Genomics (KEGG) should be cited with appropriate reference (doi:10.1093/nar/28.1.27) and all bioinformatics techniques used should be cited with appropriate reference or web link.
2. Line: 76 and 77: Mention detailed bioinformatics web resources used for this study.
3. Figure 1 and Table 1 uses reported nine datasets which should be included in reference list.
4. Figure 1: Figure legends should be changed with appropriate designation. [May use LUAD Vs. Normal Lung (Beer et al, Nat Med 2002)].
5. Data and proposed model is overstated, author should explain overall and detail results of bioinformatics analysis.
6. In Table 1 and 2: “Age at initial pathogenic diagnosis.” Entry “80” should be stated appropriately.
7. In a running text full form for OS, PPS, and RFS should be added. (Ex. OS: Overall Survival).
8. Result section needs to be redrafted with appropriate and additional details on outcome of bioinformatics analysis (Especially, for Figure 4 and 5 provide detailed explanation of analysis).
9. What are the positive and negative controls were used for data analysis of Figure 3?
10. Figure 4A: author should provide “n” values for the samples used.
11. Conclusion should be redrafted appropriately.

Overall this study deals with the data acquisition and analysis using different bioinformatics methods with initial comparison of nine datasets as shown in Figure 1. In addition to this, if author can provide additional bioinformatics analysis for recently reported dataset/s, it will be extremely important and useful for strengthening the field as well as further understanding the therapeutic role of NTK2 in LAUD.

Validity of the findings

No comments

Annotated reviews are not available for download in order to protect the identity of reviewers who chose to remain anonymous.

---

## Round 0.2 · Minor Revisions

Please incorporate the minor changes suggested by the reviewer 1. Authors have addressed all the major questions raised.

Reviewer 1 ·

Basic reporting

Overall the manuscript seems get better. The findings are worth to report and will be useful in the studies of NTKR2 expression profile. Literature is well referenced & relevant to the concept. But still here are few minor concerns that need to address before acceptace.

Experimental design

No comments

Validity of the findings

No Comments

Additional comments

It is unusual to have no space between word and reference bracket??
eg. Line: 48 treatment(Zheng et al. 2018),
Line 52: schizophrenia(Spalek et al. 2016) and many more throughout the manuscript. Author requested to check and confirm with journal guidelines.

The author should use appropriate tense to write the part of discussion in which they are discussing their current work, for eg.
Line193-195: “This study was the first to give comprehensive evidence through bioinformatics analysis of different public datasets that NTRK2 was identified as anti-oncogene in LUAD and could be used as a potential biomarker.” This is not the past work and it is been presented for the first time. It could be written as ‘This is the first study which presents comprehensive bioinformatic analysis of different public datasets that NTRK2 was identified as anti-oncogene in LUAD.’

Line 200: ‘cancer.NTRK2’ space is missing after sentence.
In ‘Acknowledgement’ section author could mention funding details. ‘Polishing’ is not an appropriate word for language editing services by commercial provider.
Present designation of the author is not required to be mentioned. Present author affiliation can be mentioned in ‘authorship and affiliation section’.

Reviewer 2 ·

Basic reporting

no comments

Experimental design

no comments

Validity of the findings

no comment

Additional comments

The manuscript: A tropomyosin receptor kinase (TRK) family protein, NTRK2 is a potential predictive biomarker for lung adenocarcinoma.
The authors have corrected the manuscript according to the suggestions of both the reviewers. They have responded well on suggestions and corrections. Therefore, I suggested this manuscript could be accepted for the publication in this journal.

---

## Round 0.3 · accepted · Accept

The authors have incorporated all the changes suggested by the reviewer.

Reviewer 1 ·

Basic reporting

The manuscript entitled " A tropomyosin receptor kinase (TRK) family protein, NTRK2 is a potential predictive biomarker for lung adenocarcinoma" by Xiang Wang et al. studies the expression profile data of NTRK2 gene in lung adenocarcinoma.

Overall the manuscript has improved after revision and can be accepted for publication.

Experimental design

NA

Validity of the findings

NA

Additional comments

Acceptable for publication.